# Modeling of Single-Process 3D-Printed Piezoelectric Sensors with Resistive Electrodes: The Low-Pass Filtering Effect

**DOI:** 10.3390/polym15010158

**Published:** 2022-12-29

**Authors:** Tilen Košir, Janko Slavič

**Affiliations:** Faculty of Mechanical Engineering, University of Ljubljana, Aškerčeva 6, 1000 Ljubljana, Slovenia

**Keywords:** low-pass filter, electrode resistance, piezoelectric sensor, material extrusion, 3D printing

## Abstract

Three-dimensional printing by material extrusion enables the production of fully functional dynamic piezoelectric sensors in a single process. Because the complete product is finished without additional processes or assembly steps, single-process manufacturing opens up new possibilities in the field of smart dynamic structures. However, due to material limitations, the 3D-printed piezoelectric sensors contain electrodes with significantly higher electrical resistance than classical piezoelectric sensors. The continuous distribution of the capacitance of the piezoelectric layer and the resistance of the electrodes results in low-pass filtering of the collected charge. Consequently, the usable frequency range of 3D-printed piezoelectric sensors is limited not only by the structural properties but also by the electrical properties. This research introduces an analytical model for determining the usable frequency range of a 3D-printed piezoelectric sensor with resistive electrodes. The model was used to determine the low-pass cutoff frequency and thus the usable frequency range of the 3D-printed piezoelectric sensor. The low-pass electrical cutoff frequency of the 3D-printed piezoelectric sensor was also experimentally investigated and good agreement was found with the analytical model. Based on this research, it is possible to design the electrical and dynamic characteristics of 3D-printed piezoelectric sensors. This research opens new possibilities for the design of future intelligent dynamic systems 3D printed in a single process.

## 1. Introduction

In recent years, the fabrication of sensory elements using additive manufacturing techniques received significant scientific attention [1,2]. The research of 3D-printed dynamic sensors is important since complex sensor shapes can be fabricated and embedded within structures already in the manufacturing stages [3]. Therefore, 3D-printed sensors make perfect candidates for applications such as structural health monitoring [4], vibration control [5], energy harvesting [6,7], metamaterials [8,9] and human health monitoring [10].

The most common sensing principles used in dynamic 3D-printed sensors include piezoresistive [11], capacitive [12,13] or piezoelectric effects [14]. While piezoresistive sensors are relatively easy to 3D print, piezoelectric sensors are much more challenging but offer much better sensitivity [14]. In the field of material extrusion (ME) and piezoelectric sensing, research is mainly focused on the fabrication of thin piezoelectric films and methods to improve their piezoelectric sensitivity [14]. Commercially available piezoelectric materials that can be used in ME include homopolymer polyvinylidene fluoride (PVDF) and its copolymer alternative PVDF-TrFe. The piezoelectric performance of thin PVDF films depends on the amount of semicrystalline β phase present in the film [14,15,16]. To achieve high sensitivity, piezoelectric films are electrically poled, with molecular dipoles aligning under a high electric field slightly below the Curie temperature [17,18].

The fabrication of thin 3D-printed piezoelectric sensors using ME usually involves the following three steps: fabrication of the film, attachment of the electrodes and electrical poling. Conventionally, poling is performed with contact electrodes [19] or corona poling [20]; however, in recent years, electric poling has also been successfully integrated into the ME process [14,21,22].

To collect the charge generated on the piezoelectric layer, electrodes must be applied to the piezoelectric layer. There are several conventional methods for electrode attachment, such as deposition [23,24], coating [25] and screen printing [26,27], all of which require multiple and different processing steps. By using ME, electrodes, piezoelectric layer and structural elements of PE sensor can all be 3D-printed in a single process [14,28]. There are several commercially available conductive filaments that can be used for the electrode deposition, such as conductive polylactic acid (PLA) [29,30], conductive thermoplastic polyurethane (TPU) [28] and conductive Electrifi filament [31]. Although there have been significant advances in polymer conductivity over the years [32], the resistivity of the most conductive filament commercially available is about $3.6\cdot 10^{-3}\;\Omega \mathrm{cm}$, which, under ideal conditions, corresponds to a resistivity about 200 times higher than copper [31]. Due to the high resistivity of conductive polymer materials used in ME, the electrode resistance can affect the sensing capabilities of 3D-printed PE sensors.

Resistive elements are used in passive shunt-connected piezoelectric transducers [33], e.g., for vibration control [34], energy harvesting [35,36] and metamaterials [37]. The effect of the continuous distribution of electrode resistance, the capacitance of the PE layer and charge generation of the PE layer were presented on the shape control of PE slender beams [38,39] with the transmission line model. In the field of PE sensing, electrode resistance of a few Ohms is not significant in the low-frequency region for most piezoelectric applications since piezoelectric transducers are frequently used in high-impedance circuits [40,41]. However, the electrodes fabricated using ME can have a resistance of several kilo-Ohms and might have a significant effect on the sensing capabilities of 3D-printed PE sensors.

In this manuscript, an analytical model and experimental method are presented to determine the usable frequency range in the same process as 3D-printed piezoelectric sensors with resistive electrodes. The presented theory is applied to a 1D piezoelectric sensor model and the usable frequency range of the sensor is determined from the electrical point of view. Then, a piezoelectric sensor with the same dimensions is 3D printed and the usable frequency range is experimentally measured and compared with the numerical results.

The manuscript is organized as follows: Section 2 describes the theoretical background, Section 3 presents the analytical model of the 3D-printed piezoelectric sensor, Section 4 describes the experimental methods used in piezoelectric sensor fabrication and measurements of electrical material properties and sensor’s impedance. Section 5 presents the measurement’s results and a comparison with the analytical model, as well as in-depth discussion, and Section 6 draws the significant conclusions.

## 2. Theoretical Background

### 2.1. Piezoelectric Effect

Using Voigt notation and the summation convention, the linear piezoelectric effect can be described by the following constitutive equations [42]:(1)$$\varepsilon_{i}=S_{ij}^{E}\sigma_{j}+d_{mi}\;E_{m}\;\;i,\;j=1,2,3,4,5,6\;m=1,2,3$$
(2)$$D_{m}=d_{mi}\;\sigma_{i}+\xi_{mk}^{\sigma }E_{k}\;\;i=1,2,3,4,5,6\;m,\;k=1,2,3$$
where $\varepsilon_{i}$ is the strain component, $S_{ij}^{E}$ is the compliance coefficient at a constant electric field, $\sigma_{i}$ is the stress component, $d_{mi}$ is the piezoelectric coefficient, $E_{m}$ is the electric field component, $D_{i}$ is the electric displacement and $\xi_{ik}^{\sigma }$ is the permittivity constant at constant stress. The poling direction of the piezoelectric layer coincides with axis 3 [43].

If piezoelectric (PE) sensors are used to measure mechanical stresses, the direct piezoelectric effect, described by Equation (Equation 2), is sufficient to study the behavior of 3D-printed PE sensors. Further, Equation (Equation 2) can be simplified, assuming PE is stressed only in the primary direction (axis 1, see Figure 1), electrodes are placed on PE film in the direction of thickness (axis 3), and in-plane electric fields $E_{2}$ and $E_{3}$ are negligible:(3)$$D_{3}=\underset{D_{3,mech}}{\underset{︸}{d_{31}\;\sigma_{1}}}+\underset{D_{3,cap}}{\underset{︸}{\xi_{33}^{\sigma }E_{3}}}$$
where $D_{3,mech}$ is electric displacement due to mechanical stress and $D_{3,cap}$ is the capacitive part of the total electric displacement [14].

### 2.2. Piezoelectric Sensing Using Charge Amplifier

One of most common approaches of measuring the charge generated on the piezoelectric film is to use charge amplifier [44]. The simplified electric circuit of an ideal piezoelectric sensor connected to the charge amplifier is shown in Figure 2: the terminals of an ideal PE sensor are connected to the cable, which is further connected to the charge amplifier. The output voltage $V_{out}\left(s\right)$ of the charge amplifier can be written as [44]:(4)$$V_{out}\left(s\right)=-\frac{1}{C_{f}}\;\frac{s\;R_{f}\;C_{f}}{1+s\;R_{f}\;C_{f}}\;\frac{1}{1+s\;R_{1}\;\left(C_{c}+C_{s}\right)}\;Q_{s}\left(s\right)$$
where $Q_{s}\left(s\right)$ is the charge generated on the PE film, $C_{s}$ is the sensor capacitance, $C_{c}$ is cable capacitance, $R_{1}$, $R_{f}$ are charge amplifier resistances and $C_{f}$ is the charge amplifier capacitance (see Figure 2). It is typically assumed that the resistances of cables and PE electrodes are negligible, which is not the case with 3D-printed sensors.

## 3. The 3D-Printed Piezoelectric Sensor Model

### 3.1. The 1D Model

The aim of this research is to simplify the modeling of 3D-printed PE sensors with resistive electrodes and enable modeling similar to an ideal PE sensor, as presented in Figure 2. The effect of electrode resistance on a 3D-printed PE sensor is researched on the simplified 1D model (see Figure 3), where it is assumed that the electric displacement due to mechanical stress $D_{3,mech}$ (Equation 3) does not vary across sensor width (axis 2). Each discrete segment is modeled as a current source in parallel with the capacitor, followed by the two resistors representing the top and bottom electrodes. Each segment’s current source $I_{i}\left(s\right)$, capacitor $C_{i}$ and resistors $R_{i}$ are modeled as [39]:(5)$$I_{i}\left(s\right)=D_{3,mech,i}\left(s\right)\;h\;b\;s=d_{31}\;\sigma_{1}\left(s\right)\;h\;b\;s$$
(6)$$C_{i}=\xi_{mk}^{\sigma }\;\frac{h\;b}{t_{p}}=\varepsilon_{r}\;\varepsilon_{0}\;\frac{h\;b}{t_{p}}$$
(7)$$R_{i}=\rho_{e}\;\frac{h}{b\;t_{e}}$$
where *b* is sensor’s width, *h* is segment’s length, *s* is Laplace complex variable, $\varepsilon_{r}$ is relative dielectric constant of piezoelectric film, $\varepsilon_{0}$ is permittivity of free space, $t_{p}$ is thickness of piezoelectric film, $\rho_{e}$ is electrode resistivity and $t_{e}$ is electrode thickness. Equidistant discretization is assumed. Using modified nodal analysis [45], each individual segment, shown in Figure 3, can be presented with:(8)$$Y_{i}\;v_{i}=I_{i}$$
(9)$$Y_{i}=\left[\begin{array}{cccc} \frac{1}{R_{i}}+s\;C_{i} & -s\;C_{i} & -\frac{1}{R_{i}} & 0\\ -s\;C_{i} & \frac{1}{R_{i}}+s\;C_{i} & 0 & -\frac{1}{R_{i}}\\ -\frac{1}{R_{i}} & 0 & \frac{1}{R_{i}} & 0\\ 0 & -\frac{1}{R_{i}} & 0 & \frac{1}{R_{i}} \end{array}\right]$$
(10)$$v_{i}=\left[\begin{array}{c} v_{i,0}\\ v_{i,1}\\ v_{i,2}\\ v_{i,3} \end{array}\right]$$
(11)$$I_{i}=\left[\begin{array}{c} I_{i}\left(s\right)\\ -I_{i}\left(s\right)\\ 0\\ 0 \end{array}\right]$$
where $Y_{i}$ is the admittance matrix, $v_{i}$ is the vector of voltage potentials and $I_{i}$ is the current vector (see Figure 3). $Y_{i}$, $v_{i}$ and $I_{i}$ for each segment (in total *N* segments) are assembled in the global admittance matrix Y, the potential vector v and the current vector I, representing the 1D model of the piezoelectric sensor:(12)Yv=I

### 3.2. Equivalent Piezoelectric Sensor Model

To simplify modeling the 3D-printed PE sensors, the 1D PE sensor model shown in Figure 4a) is simplified to an equivalent PE sensor model with equivalent current source and capacitance, as shown in Figure 4c). To obtain the equivalent PE sensor model, the Thevenin’s equivalent electric circuit [46] across the electrode terminals has to be obtained first. This is done by grounding terminal B (see Figure 4) and computing the impedance matrix Z as: (13)$$Z=Y^{\prime^{-1}}$$
where $Y^{\prime }$ is the global admittance matrix Y after ground boundary condition at terminal *B* has been taken into account (degree of freedom (DOF) of terminal *B* has been removed). The equivalent impedance $Z_{eq}\left(s\right)$ across terminals *A* and *B* is calculated as [45]:(14)$$Z_{eq}\left(s\right)=Z_{AA}\left(s\right)+Z_{BB}\left(s\right)-Z_{AB}\left(s\right)-Z_{BA}\left(s\right)=Z_{AA}\left(s\right)$$
where $Z_{AB}\left(s\right)$ is an element of global impedance matrix Z and *A*, *B* are corresponding indices of DOFs associated with PE sensor terminals *A* and *B* (see Figure 4). Since DOF B has been eliminated, it follows that $Z_{BB}\left(s\right)=Z_{AB}\left(s\right)=Z_{BA}\left(s\right)=0$. The equivalent Thevenin’s voltage source can be obtained after the vector of voltage potentials is calculated:(15)$$v^{\prime }=Z\;I^{\prime }$$
where $v^{\prime }$, $I^{\prime }$ represent global voltage potential and current vectors after the ground boundary condition at B has been taken into account. The equivalent Thevenin’s voltage $V_{th}\left(s\right)$ is then calculated as:(16)$$V_{th}\left(s\right)=v_{A}\left(s\right)-v_{B}\left(s\right)=v_{A}\left(s\right)$$
where $v_{A}\left(s\right)$ and $v_{B}\left(s\right)$ are terminal voltage potentials at *A* and *B*, respectively. The equivalent electric current $I_{eq}\left(s\right)$ and capacitance $C_{eq}\left(s\right)$ can be calculated in the next step as:(17)$$I_{eq}\left(s\right)=\frac{V_{th}\left(s\right)}{Z_{eq}\left(s\right)}$$
(18)$$C_{eq}\left(s\right)=\frac{1}{s\;Z_{eq}\left(s\right)}$$

From the stand point of determining the sensor’s usable frequency range, H(s) is defined as the ratio between the equivalent electric current $I_{eq}\left(s\right)$ and the total electric current $I_{mech}\left(s\right)$ generated by the mechanical stresses:(19)$$I_{mech}\left(s\right)=\sum_{i=1}^{N}I_{i}\left(s\right)$$
(20)$$H\left(s\right)=\frac{I_{eq}\left(s\right)}{I_{mech}\left(s\right)}=\frac{V_{th}\left(s\right)}{Z_{eq}\left(s\right)\;\sum_{i=1}^{N}I_{i}\left(s\right)}$$

Alternatively, H(s) can also be defined as the ratio between equivalent electric charge $Q_{eq}\left(s\right)$ and total electric charge generated by the mechanical stresses $Q_{mech}\left(s\right)$:(21)$$H\left(s\right)=\frac{Q_{eq}\left(s\right)}{Q_{mech}\left(s\right)}$$
where $I_{eq}\left(s\right)=s\;Q_{eq}\left(s\right)$ and $I_{mech}\left(s\right)=s\;Q_{mech}\left(s\right)$. By substituting $Q_{s}\left(s\right)=Q_{eq}\left(s\right)=Q_{mech}\left(s\right)\cdot H\left(s\right)$ and $C_{s}=C_{eq}\left(s\right)$ into Equation (Equation 4), the output voltage $V_{out}$ of the charge amplifier when using the PE 3D-printed sensor is:(22)$$V_{out}\left(s\right)=-\frac{1}{C_{f}}\;\frac{s\;R_{f}\;C_{f}}{1+s\;R_{f}\;C_{f}}\;\frac{1}{1+s\;R_{1}\;(C_{c}+\underset{C_{s}}{\underset{︸}{C_{eq}\left(s\right)}})}\;\underset{Q_{s}\left(s\right)}{\underset{︸}{H\left(s\right)\;Q_{mech}\left(s\right)}}$$

### 3.3. Special Case for H(s) and $C_{eq}\left(s\right)$ of 3D-Printed Piezoelectric Sensor

In further sections of the manuscript, emphasis is placed on a special case in which the amplitude of the electric displacement $D_{3,mech}\left(s\right)$ is constant over the entire area of the PE layer. This is true for all cases where the amplitudes of the normal mechanical stresses $\sigma_{1}$, $\sigma_{2}$ and $\sigma_{3}$ are constant over the entire PE layer area.

For the special case presented, if the system of equations given by Equation (Equation 15) is solved symbolically, it can be shown that regardless of the discretization *N*, the Thevenin’s voltage $V_{th}$ across the electrode terminals is:(23)$$V_{th}\left(s\right)=\frac{I_{mech}\left(s\right)}{s\;\sum_{i}^{N}C_{i}}=\frac{I_{mech}\left(s\right)}{s\;C_{tot}}$$
where $C_{tot}=\sum_{i}^{N}C_{i}$ represents the total 3D-printed sensor capacitance. By using Equations (Equation 18), (Equation 20) and (Equation 23), H(s) can be simplified to:(24)$$H\left(s\right)=\frac{I_{eq}\left(s\right)}{I_{mech}\left(s\right)}=\frac{V_{th}\left(s\right)}{Z_{eq}\left(s\right)\;I_{mech}\left(s\right)}=\frac{I_{mech}\left(s\right)\;C_{eq}\left(s\right)\;s}{I_{mech}\left(s\right)\;C_{tot}\;s}=\frac{C_{eq}\left(s\right)}{C_{tot}}=\frac{Z_{C}\left(s\right)}{Z_{eq}\left(s\right)}$$
(25)$$Z_{C}=\frac{1}{s\;C_{tot}}$$
where $Z_{C}\left(s\right)$ is the impedance of an ideal PE sensor with negligible electrode resistance. Equation (Equation 24) suggests that in cases where the amplitude of $D_{3,mech}\left(s\right)$ (Equation 5) is constant over the area of the PE film, H(s) can also be determined experimentally if one can measure the impedance of the 3D-printed PE sensor $Z_{eq}$ and the impedance of the ideal PE sensor $Z_{C}$ with the same dimensions.

For the special case described, Figure 5 shows H(s) and $C_{eq}\left(s\right)$ for different numbers of segments *N* using the constants: L=100mm, b=10mm, $t_{e}=t_{p}=0.2\;\mathrm{mm}$, $\rho_{e}=0.220\;\Omega m$, $\varepsilon_{r}=8.5$, $\varepsilon_{0}=8.85\cdot 10^{-12}\;m^{-3}\;{\mathrm{kg}}^{-1}\;s^{4}\;A^{2}$. The H(s) shows that the resistive electrodes of the 3D-printed PE sensor, coupled with the capacitance of the PE film, form a low-pass filter. As the number of segments *N* increases, the cutoff frequency of the sensor increases and reaches convergence at about N=100. Moreover, the equivalent sensor capacitance $C_{eq}\left(s\right)$ is constant in the low-frequency range (approximately below 5kHz) and the measured charge is equal to the generated charge since |H(s)|≈1. It follows that the 3D-printed PE sensor can be treated as an ideal sensor with negligible electrode resistance if used below the cutoff frequency H(s).

As mentioned, the convergent solution is obtained at N≈100; however, the solution at N=1 is also considered to obtain a simple description of the low-pass filter. For N=1, H(s) corresponds to a traditional RC low-pass filter described as follows [47]:(26)$$H\left(s\right)=\frac{1}{1+2R_{tot}\;C_{tot}\;s}$$
(27)$$R_{tot}=\rho_{e}\frac{L}{t_{e}\;b}$$
(28)$$C_{tot}=\varepsilon_{0}\;\varepsilon_{r}\;\frac{L\;b}{t_{p}}$$
where $R_{tot}$ and $C_{tot}$ represent the total electrode resistance and total capacitance of the 3D-printed PE sensor, respectively. Therefore, the low-pass cutoff frequency $f_{c}$ for N=1 is [48]:(29)$$f_{c}=\frac{1}{2\pi }\;\frac{1}{2R_{tot}\;C_{tot}}$$

Equation (Equation 29) provides an estimate of the usable frequency range of the 3D-printed PE sensor. From Figure 5, it can be seen that the estimated cutoff frequency $f_{c}$ is about 2.6 times lower than the actual cutoff frequency. The ratio between estimated and actual cutoff frequency $f_{c}$ depends on the electrical boundary conditions (location of terminals *A* and *B*), but since the estimated cutoff frequency is always lower, Equation (Equation 29) can be used to determine the usable frequency range of 3D-printed PE sensors in the initial design stages.

## 4. Experimental Methods

### 4.1. The 3D-Printed Piezoelectric Sensor

To investigate the effect of resistive electrodes on the sensing capability of the 3D-printed PE sensor, two specimens were 3D-printed: specimen A, also referred to as the 3D-printed sensor, and specimen B, also referred to as the ideal sensor. Specimen A was fully 3D printed (including CPLA electrodes) in a single process, while specimen B has a 3D printed PE layer and manually applied electrodes. Since the resistivity of CPLA is about four orders of magnitude greater than that of silver paint, specimen B with silver paint electrodes can be considered an ideal PE sensor compared to specimen A.

Specimen A with dimensions shown in Figure 6a,b was 3D printed with the E3D Toolchanger using three different materials: polyvinylidene fluoride (PVDF) from Nile Polymers as the piezoelectric layer, conductive polylactic acid (CPLA)/carbon black from Protopasta as electrodes and Prusament polylactic acid (PLA) from Prusa for other structural elements. Each material was extruded using a separate Matrix extruder from Trianglelab to avoid mixing and contamination of the materials. Each filament was extruded with a layer height of 0.2 mm and a nozzle diameter of 0.4 mm; the other printing parameters are shown in Table A1 in Appendix A. The G-code file for material extrusion was created with Prusa Slicer 2.4.0.

Electrical contact with the copper wires was made in a manner similar to that described in [14,29]. The enameled copper wire was soldered to the 3M conductive copper tape. The copper tape was then glued to the edge of the 3D-printed electrodes, where a thin layer of conductive silver paint was also applied. All four sensor edges are electrically contacted. This allows the sensor’s impedance to be measured and the electrode resistance to be measured on the same sample. The finished 3D-printed sensor can be seen in Figure 6c.

Specimen B (see Figure 6d) was 3D printed from PVDF with the same parameters and dimensions as specimen A. The electrodes were deposited with Electrolube conductive silver paint (surface resistivity of 0.01–0.03 $\Omega /{\mathrm{cm}}^{2}$) in an additional procedure after 3D printing. Specimen B is used to measure the dielectric constant $\varepsilon_{r}$ of the 3D printed PVDF layer.

Specimens A and B were electrically poled for 45min at a constant voltage of 3.0kV using an HVDC converter (Ultra 15AV12-P4, Advanced Energy) at a temperature of 85 ${}^{\circ }C$ to obtain piezoelectric properties as described in [14].

### 4.2. Electrode Resistivity and Dielectric Constant of Piezoelectric Layer

The first objective of the experimental work was to obtain the electrical material properties necessary to calculate the H(s) (Equation 21) and the cutoff frequency. For this purpose, the electrode resistivity $\rho_{e}$ and the dielectric constant $\varepsilon_{r}$ of the PE layer were measured as described in Appendix B and Appendix C, respectively.

### 4.3. Impedance Measurement

The second objective of the experimental work was to measure the cutoff frequency of the sensor. Using the measured dielectric constant $\varepsilon_{r}$ of PVDF on specimen B, the impedance of an ideal sensor $Z_{C}$ (Equation 25) can be calculated. By measuring the impedance of the 3D-printed sensor $Z_{eq}$ (Equation 14) (specimen A) at terminals A and B (see Figure 6), H(s) (Equation 24) and the low-pass cutoff frequency can be determined for loading cases where the electric displacement $D_{3}$ has a constant amplitude over the entire area of the PE layer. Several impedance measurements were performed on specimen A at different sensor lengths *L*. The measurements at the different lengths *L* started with the 140mm long specimen, gradually decreasing by 20mm to: 120mm, 100mm, 80mm, 60mm and 40mm.

## 5. Results

### 5.1. Electrode Resistivity $\rho_{e}$ and Dielectric Constant $\varepsilon_{r}$ of Piezoelectric Layer

Table 1 contains the measured resistivity $\rho_{e}$ of silver paint and CPLA electrodes. The resistivity of CPLA is about four orders of magnitude larger than that of silver paint. For both electrode materials, the measured resistances of the top and bottom electrodes show some discrepancy. In the case of the 3D-printed electrodes, the discrepancy could be primarily due to slightly different extrusion conditions, as the lower electrode is squeezed between the build plate and nozzle during printing, resulting in more conductive paths and thus lower resistance. The resistance variations in silver paint electrodes are primarily due to the deposition method, as a perfectly uniform coating is difficult to achieve.

The measured relative dielectric constant $\varepsilon_{r}$ of the PVDF (active PE layer) versus the frequency is shown in Figure 7. A good agreement of the measured dielectric constant for different lengths of specimen B can be observed. In the low frequency range, the measurement uncertainty is increased because the reference resistors in the impedance analyzer were limited to 1MΩ. Although the active PE layer is 3D printed, the measured relative dielectric constant of PVDF is within the range of those measured in [49,50].

The measured relative dielectric constants at different specimen lengths *L* were averaged at each frequency step to obtain a dielectric constant of PVDF as a function of frequency (shown by the black dashed curve in Figure 7). The measured resistivity $\rho_{e}$ of CPLA and dielectric constant $\varepsilon_{r}$ of PVDF are used to calculate $Z_{eq}$ (Equation 14), $C_{eq}$ (Equation 18), and H(s) (Equation 20) of the 3D-printed PE sensor.

### 5.2. The 3D-Printed Sensor Impedance $Z_{eq}$

Using the measured dielectric constant of PVDF $\varepsilon_{r}$, the resistivity of CPLA $\rho_{e}$, and the geometry of the 3D-printed PE sensor (specimen A), the impedance $Z_{eq}$ (Equation 14) of the PE 3D-printed sensor is calculated (discretization N=100 is used). Figure 8 shows the directly measured and calculated impedances $Z_{eq}$ of the 3D-printed PE sensor at different lengths *L*. The amplitude spectrum (Figure 8a) shows good agreement between measured and calculated sensor impedance over the entire frequency range. The phase spectrum shows a reasonable discrepancy (indicated by the red arrow in Figure 8b) between the numerical model and measurement at higher frequencies (i.e., above 100kHz for L=140mm). The differences could be due to a possibly slightly different dielectric loss of PVDF between specimens A (3D-printed PE sensor) and B (ideal PE sensor), since the PVDF dielectric constant used in the calculations is from specimen B. At lower frequencies (i.e., below 100kHz for L=140mm), both amplitude and phase spectra of the 3D-printed PE sensor show good agreement between numerical model and measurement.

### 5.3. H(s) and the Cutoff Frequency

Using Equations (Equation 18) and (Equation 24), $C_{eq}\left(s\right)$ and H(s) are calculated based on the measured electrical material properties of the 3D-printed PE sensor. In addition, H(s) and $C_{eq}\left(s\right)$ are also determined based on the measured impedance $Z_{eq}$ of the 3D-printed PE sensor and the measured dielectric constant $\varepsilon_{r}$, which is used to calculate the impedance of an ideal PE sensor $Z_{C}$. The results are shown in Figure 9.

The amplitude spectra of H(s) and $C_{eq}\left(s\right)$ (Figure 9a,b) show good agreement between the analytical model and the measurement. Larger specimen lengths generally show better agreement in the cutoff frequency region. This is due to the fact that the area contacted with silver paint and conductive tape (see Figure 6) increases the conductance of the electrode at the sensor edge. This effect is larger for shorter specimens, where the total resistance of the electrode in the measurement is lower than in the numerical model. The latter is also evident from the slightly higher measured cutoff frequency compared to the calculated one.

At higher frequencies above the cutoff frequency, a discrepancy in phase angle between the analytical model and the measurement is observed. As discussed in Section 5.2, the discrepancy can be explained by a possible difference in the dielectric loss of PVDF between specimens A and B. From the point of view of the use and operation of the 3D-printed PE sensor, the phase angle deviation in the frequency range above the cutoff frequency is not significant because the expected use of the 3D-printed sensor is far below the cutoff frequency.

The measured and calculated cutoff frequencies (at 3dB drop) for different specimen lengths are shown in Table 2. A higher cutoff frequency is observed for shorter specimen lengths *L*. The resistance between each material point where the charge is generated and each terminal where the charge is collected, is on average, lower for shorter specimen lengths, which is why a higher cutoff frequency is observed for shorter specimen lengths. This is to be expected since lower resistance leads to a higher cutoff frequency. Table 2 also contains the estimated cutoff frequencies obtained from Equation (Equation 29). It can be observed that for each specimen length *L* the estimated cutoff frequency is lower than the measured one and the estimate is therefore on the safe side.

Overall, good agreement is observed between the analytical model and the measurement in the frequency range below the 3dB drop cut-off frequency, where the 3D-printed sensors are expected to be used. Therefore, the presented measurement and modeling methods can be used to determine the usable frequency range of 3D-printed PE sensors from an electrical perspective. Further, if the 3D-printed sensor is connected to a charge amplifier, an ideal PE sensor with negligible resistance can be assumed in the operating frequency range significantly below the cutoff frequency.

## 6. Conclusions

Classical piezoelectric sensors have electrodes with negligible resistivity; this is not the case in the recently introduced 3D printed piezoelectric sensors. This research introduces an analytical model for determining the usable frequency range of 3D-printed piezoelectric sensors with resistive electrodes. The 1D model of a 3D-printed piezoelectric (PE) sensor, where the PE sensor was divided into several smaller segments, with each segment modeled as a current source and capacitor in parallel (PE layer), followed by two resistors (resistive electrodes). The 3D-printed 1D PE sensor model is then simplified to a PE sensor model with equivalent current source and equivalent capacitance. Here, the equivalent capacitance $C_{eq}\left(s\right)$ is defined as well as the transfer function H(s) between the collected charge and the total charge generated on the PE layer.

Based on the measured impedance $Z_{eq}\left(s\right)$ of the 3D-printed piezoelectric sensor and the measured electrical properties of the materials, such as the resistivity of the electrodes and the dielectric constant of the piezoelectric layer, the charge transfer function H(s) and the equivalent capacitance $C_{eq}\left(s\right)$ of the 3D-printed PE sensor were also determined experimentally and compared with the analytical model. In the frequency range below the cutoff frequency, all specimen lengths with different active surface areas showed good agreement between measured and calculated H(s) and $C_{eq}\left(s\right)$, while in the cutoff frequency range, longer specimen lengths (larger surface area) generally showed better agreement.

From H(s), a low-pass cutoff frequency and thus the usable frequency range of the 3D-printed PE sensor can be determined. For the given different specimen dimensions, cutoff frequencies between 25.2kHz and 309kHz were calculated and cutoff frequencies between 24.5kHz and 403kHz were measured. A lower cutoff frequency was measured for longer specimen lengths (larger active surface area). In the frequency range significantly below the cutoff frequency, the 3D-printed PE sensor can be modeled as an ideal PE sensor with negligible electrode resistance connected to a charge amplifier. The introduced model is not limited to 1D cases and can be applied to 2D or 3D numerical models without modifications, except for the derivation of the electrical admittance matrix of the PE sensor.

## Figures and Tables

**Figure 1 polymers-15-00158-f001:**
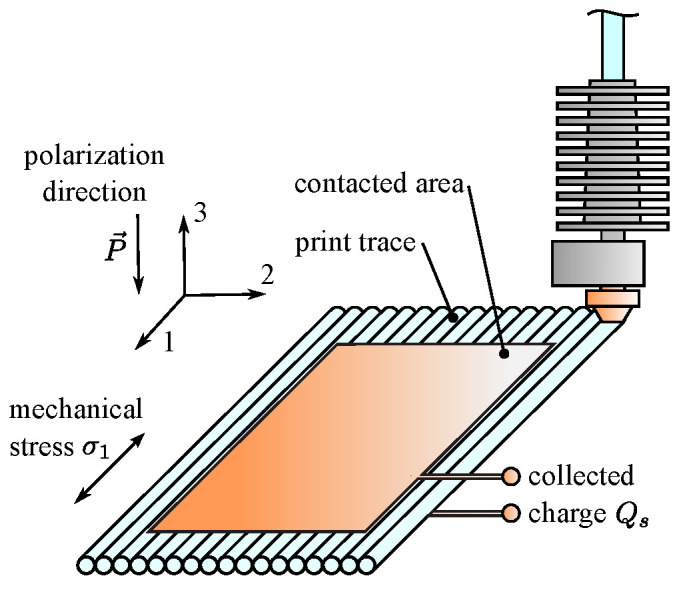
Coordinate system for a piezoelectric PVDF film fabricated with ME.

**Figure 2 polymers-15-00158-f002:**
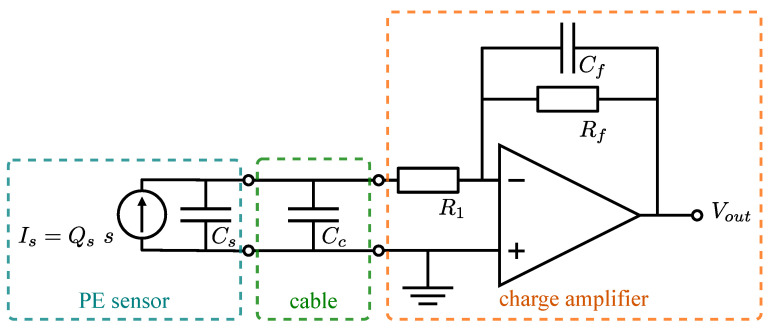
Electric circuit of piezoelectric sensor connected to charge amplifier.

**Figure 3 polymers-15-00158-f003:**
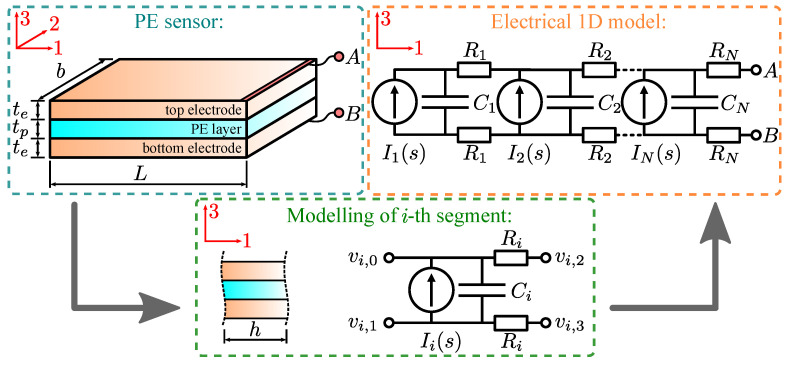
The 1D electrical model of the piezoelectric 3D printed sensor.

**Figure 4 polymers-15-00158-f004:**
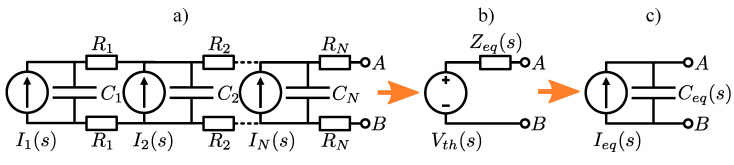
(**a**) The 1D PE sensor model with resistive electrodes. (**b**) PE sensor representation with Thevenin’s equivalent electric circuit. (**c**) PE sensor model with equivalent current source $I_{eq}$ and capacitance $C_{eq}$.

**Figure 5 polymers-15-00158-f005:**
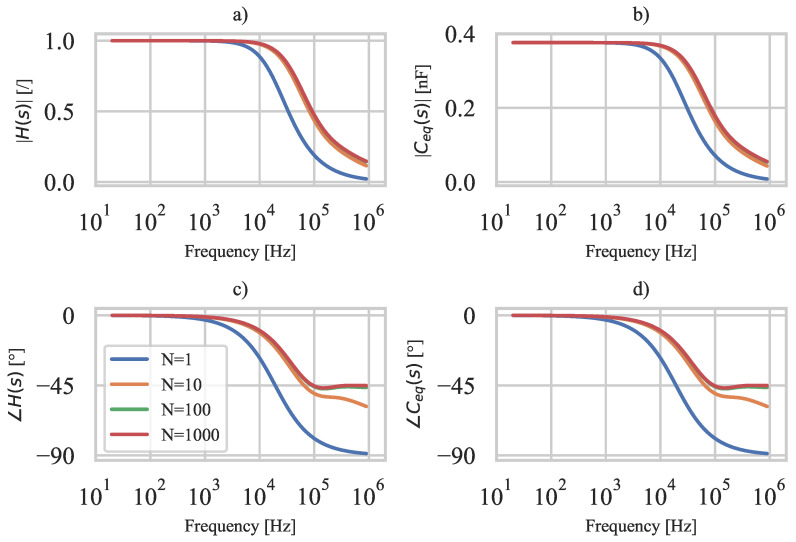
The 3D-printed sensor characteristics for different *N*: (**a**) H(s) amplitude spectrum, (**b**) $C_{eq}\left(s\right)$ amplitude spectrum, (**c**) H(s) phase spectrum, (**d**) $C_{eq}\left(s\right)$ phase spectrum.

**Figure 6 polymers-15-00158-f006:**
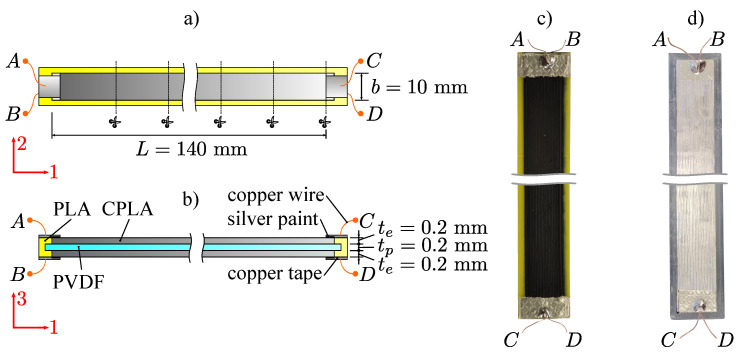
Three-dimensionally-printed specimens A and B: (**a**) top view, (**b**) side view, (**c**) 3D-printed PE sensor (specimen A), (**d**) ideal PE sensor (specimen B).

**Figure 7 polymers-15-00158-f007:**
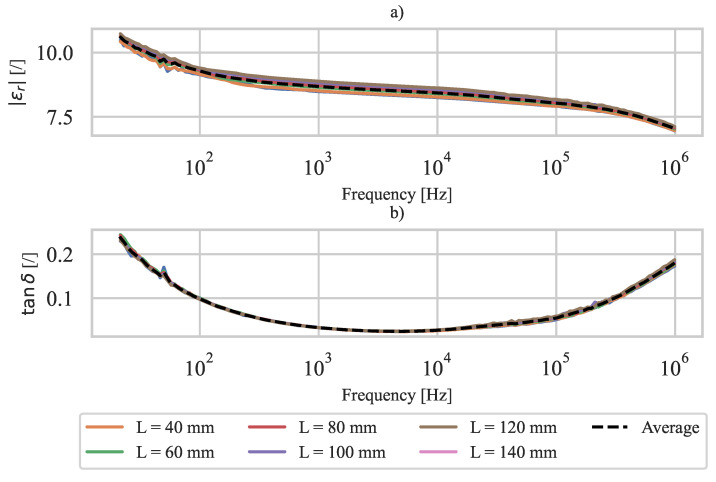
Measured dielectric constant $\varepsilon_{r}$ of PVDF: (**a**) absolute dielectric constant, (**b**) dielectric loss.

**Figure 8 polymers-15-00158-f008:**
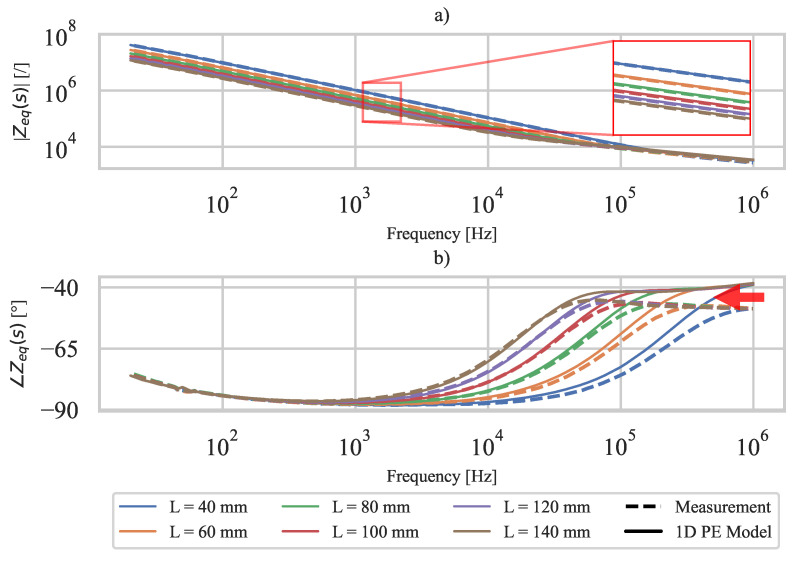
Measured and calculated sensor impedance $Z_{eq}$: (**a**) amplitude spectrum, (**b**) phase spectrum.

**Figure 9 polymers-15-00158-f009:**
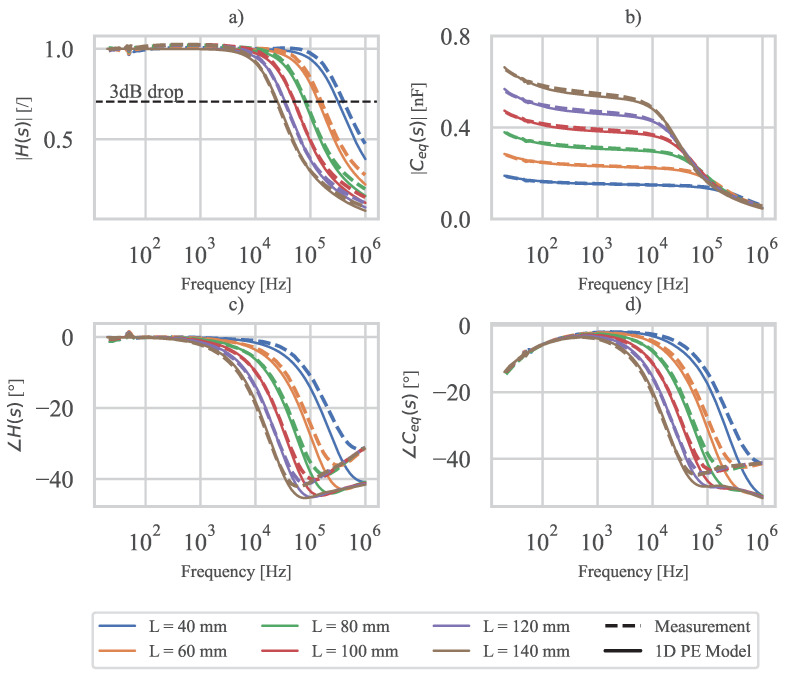
Measurement and 1D PE model comparison: (**a**) H(s) amplitude spectrum, (**b**) $C_{eq}\left(s\right)$ amplitude spectrum, (**c**) H(s) phase spectrum, (**d**) $C_{eq}\left(s\right)$ phase spectrum.

**Table 1 polymers-15-00158-t001:** Measured resistances of CPLA and silver paint electrodes and calculated average resistivities $\rho_{e}$.

Material	Top Electrode Resistance (Ω)	Bottom Electrode Resistance (Ω)	Average Resistivity (Ω m)
CPLA	16,832	15,873	0.220
silver paint	1.58	2.70	$2.88\times 10^{-5}$

**Table 2 polymers-15-00158-t002:** Measured, calculated and estimated low-pass cutoff frequencies for different 3D-printed PE sensor lengths.

Specimen Length (mm)	Measured Cutoff Freq. (Hz)	Calculated Cutoff Freq. (Hz)	Estimated Cutoff Freq. (Hz)
40	403,389	309,007	119,869
60	157,073	137,337	53,276
80	82,535	77,252	29,967
100	50,882	49,441	19,179
120	33,977	34,334	13,319
140	24,520	25,225	9785

## Data Availability

Not applicable.

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
