# Peer review of "Modeling of Single-Process 3D-Printed Piezoelectric Sensors with Resistive Electrodes: The Low-Pass Filtering Effect"

_polymers, 2022, doi:10.3390/polym15010158_

Round 1

Reviewer 1 Report

I have reviewed the paper "Modeling of single-process 3D-printed piezoelectric sensors with resistive electrodes" and found the paper can be accepted after major revision.

-More physical explanation of results is required.

-The Abstract should be improved.

-The quality of the figures should be improved.

-Finally, the language of the paper needs to be polished.

-Based on the topic the title is so short and needs to be clarify the problem statement clearly.

-The Figures quality are too weak please improve the quality and put some arrays on the important part

-The language of the paper needs major polish.

The following reference are introduced to compare for the preparation by 3D print. 

Karimi, M., Asefnejad, A., Aflaki, D., Surendar, A., Baharifar, H., Saber-Samandari, S., ... & Toghraie, D. (2021). Fabrication of shapeless scaffolds reinforced with baghdadite-magnetite nanoparticles using a 3D printer and freeze-drying technique. Journal of Materials Research and Technology14, 3070-3079.

Author Response

The authors are thankful to the reviewer for his/her effort.

Reviewer: I have reviewed the paper "Modeling of single-process 3D-printed piezoelectric sensors with resistive electrodes" and found the paper can be accepted after major revision.

------------------------------------------------------------------
Reviewer 1: More physical explanation of results is required.

The authors thank the reviewer for pointing this out. Upon further review of results and additional explanation of the results in section 5.3 was added.

Additional explanation of the results added:
"At higher frequencies above the cutoff frequency, a discrepancy in phase angle between the analytical model and the measurement is observed. As discussed Sec. 5.2, the discrepancy can be explained by a possible difference in the dielectric loss of PVDF between specimens A and B. From the point of view of the use and operation of the 3D-printed PE sensor, the phase angle deviation in the frequency range above the cutoff frequency is not significant because the expected use of the 3D-printed sensor is far below the cutoff frequency."

------------------------------------------------------------------
Reviewer 1: The Abstract should be improved.

The authors are grateful for the comment provided by the reviewer. The abstract was revised and English was improved by the native English speaker. The changes made are highlighted with "blue" color. 

------------------------------------------------------------------
Reviewer 1: The quality of the figures should be improved.
Reviewer 1:  The Figures quality are too weak please improve the quality and put some arrays on the important part

The authors thank the reviewer for the comment. The figures were replaced with vector images to improve their quality. Figure 8 was supplemented with an red arrow indicating the phase discrepancies discussed in Section 5.2 and red arrow was mentioned in text (line 211) to improve research clarity. Additionally, a horizontal dashed line was added to Figure 9 a) to indicate the 3dB drop where the cutoff frequency is defined.

------------------------------------------------------------------

Reviewer 1: Finally, the language of the paper needs to be polished.
Reviewer 1: The language of the paper needs major polish.

The authors thank the reviewer for the comment. The research paper was proof-read by a native English speaker and English used in the paper should now be on a higher level.

------------------------------------------------------------------
Reviewer 1: Based on the topic the title is so short and needs to be clarify the problem statement clearly.

The authors thank the reviewer for this valuable insight. The title of the article was reviewed and extended to "Modeling of single-process 3D-printed piezoelectric sensors with resistive electrodes: the low-pass filtering effect"

------------------------------------------------------------------
Reviewer 1: The following reference are introduced to compare for the preparation by 3D print. 
Karimi, M., Asefnejad, A., Aflaki, D., Surendar, A., Baharifar, H., Saber-Samandari, S., ... & Toghraie, D. (2021). Fabrication of shapeless scaffolds reinforced with baghdadite-magnetite nanoparticles using a 3D printer and freeze-drying technique. Journal of Materials Research and Technology, 14, 3070-3079.

Upon reading the suggested research, the authors cited the work in the introduction section, where different materials used to fabricate piezoelectric sensors are presented and discussed (4th paragraph).

Reviewer 2 Report

This manuscript describes an analytical model for determining the range of frequency employed in 3D-printed piezoelectric sensors with resistive electrodes. The manuscript presents with sufficient clarity the problem to be addressed, and the results obtained are certainly interesting and worthy of being brought to the attention of the readers of this journal involved in these technological problems. 

I have no particular criticism or comments to make to the authors although the introduction section could be shortened a little and perhaps emphasize the results obtained a little more.

A few minor suggestions

Fig. 7 adds nothing of information and could be eliminated

The sentence at lines 237-241 is unclear and the deviations observed at higher frequencies should be better explained

Author Response

The authors are thankful to the reviewer for his/her effort.

This manuscript describes an analytical model for determining the range of frequency employed in 3D-printed piezoelectric sensors with resistive electrodes. The manuscript presents with sufficient clarity the problem to be addressed, and the results obtained are certainly interesting and worthy of being brought to the attention of the readers of this journal involved in these technological problems. 

------------------------------------------------------------------
Reviewer 2: I have no particular criticism or comments to make to the authors although the introduction section could be shortened a little and perhaps emphasize the results obtained a little more.

The authors thank the reviewer for the comment. The introduction was revised and shortened where possible. The rewritten parts in the introduction are highlighted with "green" color.

The results were discussed in greater detailes in section 5.3 by including additional comments suggested in the last feedback comment provided by the reviewer (please see our response to your last comment.)

------------------------------------------------------------------
Reviewer 2: Fig. 7 adds nothing of information and could be eliminated

The authors thank the reviewer for this valuable comment. Figure 7 was removed from the research paper.

------------------------------------------------------------------
Reviewer 2: The sentence in lines 237-241 is unclear and the deviations observed at higher frequencies should be better explained

The authors thank the reviewer for addressing his comment about the result section. Sentence 237-241 was revised and rewritten to: "The resistance between each material point where the charge is generated and each terminal where the charge is collected, is on average lower for shorter specimen lengths, which is why a higher cutoff frequency is observed for shorter specimen lengths. This is to be expected since lower resistance leads to a higher cutoff frequency." - see the text highlighted with "green" color in the results section 5.3.

The authors also added additional comments regarding the phase angle discrepancies between the analytical model and the measurements. The following paragraph was added: "At higher frequencies above the cutoff frequency, a discrepancy in phase angle between the analytical model and the measurement is observed. As discussed Sec. 5.2, the discrepancy can be explained by a possible difference in the dielectric loss of PVDF between specimens A and B. From the point of view of the use and operation of the 3D-printed PE sensor, the phase angle deviation in the frequency range above the cutoff frequency is not significant because the expected use of the 3D-printed sensor is far below the cutoff frequency." - please see the text highlighted with "blue" color in section 5.3